# Unravelling the Role of Glycogen Synthase Kinase-3 in Alzheimer’s Disease-Related Epileptic Seizures

**DOI:** 10.3390/ijms21103676

**Published:** 2020-05-23

**Authors:** Runxuan Lin, Nigel Charles Jones, Patrick Kwan

**Affiliations:** 1Department of Neuroscience, Central Clinical School, Monash University, Melbourne, Victoria 3004, Australia; runxuan.lin@monash.edu; 2Department of Neurology, Alfred Health, Melbourne, Victoria 3004, Australia

**Keywords:** Glycogen synthase kinase 3, Alzheimer’s disease, Epileptic seizures, Hyperphosphorylated tau, Amyloid beta

## Abstract

Alzheimer’s disease (AD) is the most common form of dementia. An increasing body of evidence describes an elevated incidence of epilepsy in patients with AD, and many transgenic animal models of AD also exhibit seizures and susceptibility to epilepsy. However, the biological mechanisms that underlie the occurrence of seizure or increased susceptibility to seizures in AD is unknown. Glycogen synthase kinase-3 (GSK-3) is a serine/threonine kinase that regulates various cellular signaling pathways, and plays a crucial role in the pathogenesis of AD. It has been suggested that GSK-3 might be a key factor that drives epileptogenesis in AD by interacting with the pathological hallmarks of AD, amyloid precursor protein (APP) and tau. Furthermore, seizures may also contribute to the progression of AD through GSK-3. In this way, GSK-3 might be involved in initiating a vicious cycle between AD and seizures. This review aims to summarise the possible role of GSK-3 in the link between AD and seizures. Understanding the role of GSK-3 in AD-associated seizures and epilepsy may help researchers develop new therapeutic approach that can manage seizure and epilepsy in AD patients as well as decelerate the progression of AD.

## 1. Introduction

Alzheimer’s disease (AD) is the most common form of dementia and is clinically characterised by cognitive impairment and progressive loss of memory. It has been estimated that one in every 85 individuals worldwide will be living with AD by 2050 [1], imposing a tremendous psychological and financial burden on the patient’s family and the community. There is currently no cure for AD, and treatments are mostly symptomatic [2].

Recent evidence has raised the possibility of a connection between AD and epilepsy, another common CNS disorder in the elderly that is characterised by an enduring predisposition to generate unprovoked seizures. Multiple studies have reported an increased incidence of unprovoked seizures in AD patients [3,4,5]. For example, prospective studies showed that AD patients are 6- to 10-fold more likely to develop seizures than age-matched controls [4,5], while it has been estimated that 10%–17% of AD patients have unprovoked seizures, suggesting that AD itself might be a risk factor of epilepsy [3,6]. Additionally, it is reported that unprovoked seizures might aggravate cognitive deficits of AD patients [7,8], which suggests that recurrent seizures lead to a vicious cycle of cognition exacerbation in AD patients.

In addition to the clinical evidence, animal models harbouring AD-associated mutations provide a useful insight on the possible causation and mechanisms underlying AD-associated epilepsy [9]. There are two main pathological hallmarks in AD, senile plaques and neurofibrillary tangles (NFT), the major molecular constituents of which are the amyloid precursor protein (APP) and microtubule-associated protein tau (MAPT), respectively [10]. Several lines of evidence describe AD-relevant mouse models, particularly those carrying APP or tau mutations, as being more susceptible of developing unprovoked seizures and induced seizures [11,12,13,14]. Others have shown that the increase of seizure susceptibility in APP mouse model was dependent on the expression of tau [15,16], which suggests that intervening the interaction between amyloid and tau might have therapeutic potential for epilepsy management in AD patients. Therefore, it is important to clarify the interaction between APP and tau and their implication in AD-associated seizures.

Glycogen synthase kinase-3 (GSK-3) is a proline-rich serine/threonine kinase which plays a pivotal role in the pathogenesis of AD [17,18,19]. It is suggested that GSK-3 may be a critical connecting factor between amyloid and tau [20,21,22], as amyloid activates GSK-3, which subsequently phosphorylates tau, the process of which is shown in Figure 1. Since the occurrence of seizures might be related to the interaction between amyloid and tau, it is likely that GSK-3 may play a unique role in the development of unprovoked seizures in AD. Therefore, a better understanding of the role of GSK-3 in connecting AD and seizures would be helpful for understanding causal mechanisms and for improving seizure management in AD patients.

In this article, we will review the possible mechanisms of AD-related seizures, present hypothesised pathways of the role of GSK-3 in the development of seizures in AD, and propose future research to clarify the role of GSK-3 in this process. It is worth noting that multiple studies using AD animal models employed models that harbour familial AD mutations and, thus, the neuropathological process in the context of familial AD. However, that does not exclude patients with sporadic AD from the possibility of similar pathological changes.

## 2. AD-Associated Epilepsy and Epileptic Seizures

There is growing clinical evidence suggestive of an increased risk of epilepsy or epileptic seizures in AD. Higher incidence of unprovoked seizures in AD patients has been reported by numerous studies [3,23,24,25]. It has been shown that seizure-free AD patients are more likely to develop unprovoked seizures compared to age-match controls [4,6]. Furthermore, evidence showed that familial AD was strongly associated with epileptogenesis [25,26], indicating that some AD-related mutations might contribute to abnormal neuronal network functions that lead to the development of epilepsy. However, it remains unclear whether seizure is an integral part of AD. There are a few confounding factors that need to be considered. On the one hand, the causal relationship is far from conclusive as the onset of seizures might merely be concurrent with severe AD or consequential to AD symptoms instead of AD per se. It is reported that about 22% of patients with dementia might conduct self-injurious behaviors [27]. Poor judgement and impaired consciousness can also lead to epilepsy-causing injuries in AD patients. On the other hand, it is also possible that the incidence of seizures is underestimated. AD patients might develop nonconvulsive seizures that can be misdiagnosed as AD symptoms such as altered consciousness or amnestic wandering [8]. Using long-term electroencephalogram (EEG) recording, clinically silent seizures were detected in AD patients [28,29], indicating that subtle seizures might not be documented in some AD patients without routine EEG recording. Therefore, the causal relationship and the underlying mechanism between AD and epilepsy need to be further clarified in prospective clinical studies.

In parallel to the clinical evidence, many transgenic mouse models carrying identified AD mutations also show unprovoked seizures or epileptiform EEG patterns. Although AD mouse models cannot fully recapitulate the pathological features of AD, they serve as controllable models of the disease with one or few genetic modifications that manifest AD-associated symptoms. The most well-studied models are mice that overexpress human mutant APP or specific APP cleavage products such as amyloid intracellular domain (AICD). The process of APP cleavage is shown in Figure 1A. Several APP models with various mutations exhibit spontaneous seizures or epileptiform discharges [13,14,30,31,32], while the genetic suppression of APP can abate epileptiform activities in APP transgenic mice [33], suggesting that APP is involved in the development of seizure. Furthermore, transgenic mice overexpressing AICD, the intracellular domain of APP, also exhibit EEG abnormalities and increased susceptibility to induced seizure [34]. In addition, APP transgenic models with a C-terminus cleavage site mutation which prevents the cleavage of AICD showed a complete reversal of AD phenotypes susceptibility to induced-seizure [34,35,36], suggesting that AICD might be the excitotoxic fragment of APP that contributes to the development of AD phenotypes and seizures in AD models. Overall, a vast literature documents consistent evidence that mutated or overexpressed APP or its cleavage products can result in seizures.

Aside from the potential immediate consequences of epileptic seizures, such as physical injuries, to AD patients, it is found that the occurrence of seizures might also be relevant to the acceleration of cognition impairment, which creates a complex bidirectional association between AD and seizures. The cognitive functions of patients with AD or other dementias may abruptly worsen after the first onset of seizure [37,38], suggesting a short-term impact on cognitive deficit. In addition, it has also been reported that AD patients with seizures had an earlier onset of cognitive decline and worse cognitive outcome compared to AD patients without seizures [30,39,40], suggesting that seizures might also have long-term effects on cognition. Similar to the clinical observation, studies on animal models also showed worsened cognitive conditions in animals with seizures [39]. Further studies suggested that epileptiform activity might disrupt memory consolidation and cause cognitive impairment in animal models [30,40,41]. Strikingly, multiple lines of evidence showed that antiepileptic drugs (AEDs), which can prevent these seizures and/or epileptiform discharges, seem capable of rescuing the cognitive deficit in AD models [42,43,44,45]. These studies highlight the significance of seizures and epilepsy in the development of AD, which also points to a potentially new therapeutic strategy for decelerating the cognitive deficit in AD patients.

## 3. GSK-3 Is Involved in the Development of Seizures in AD

GSK-3 is a ubiquitously expressed kinase that is involved in various physiological and pathological processes. GSK-3 was originally found as a key kinase that phosphorylated glycogen synthase during glycogen accumulation, a fundamental process of energy storage [46]. There are two genes coding for the GSK-3 protein, from which two homologous forms are expressed, GSK-3α and GSK-3β. Similar to most kinases, the activity of GSK-3 is regulated by phosphorylation at certain sites. The inhibition of GSK-3 is regulated by phosphorylation at serine residue near N-terminus (Ser9 of GSK-3β and Ser21 of GSK-3α) [47]. When GSK-3 is activated, the tyrosine site of GSK-3 will be auto-phosphorylated (Tyr216 of GSK-3β and Tyr279 of GSK-3α) [48]. Although GSK-3α shares 98% sequence identity with GSK-3β [49], most studies have focused on GSK-3β and the functions of GSK-3α are not well understood. It is possible that GSK-3β serves a more fundamental physiological function because the GSK-3β knockout is lethal [50], whereas knockout of GSK-3α is tolerable [51]. GSK-3 is involved in multiple molecular events including cell signalling [52], maintaining cellular structure [53], and modulation of transcription [52]. Other studies suggested that GSK-3 might also be involved in epileptogenesis [54].

The role of GSK-3 in the pathogenesis of AD has been discussed in recent reviews [17,55,56,57]. GSK-3 was initially known as one of the key kinases that was involved in the phosphorylation of tau [46,58,59], which is a biological process that regulates the binding of tau to microtubules, as being shown in Figure 1C [60]. However, GSK-3 is excessively activated in AD, which contributes to the hyper-phosphorylation of tau [60]. Hyper-phosphorylated tau has been identified as one of the critical factors and causal drivers of pathogenesis of AD [61,62]. Furthermore, studies showed that hyper-phosphorylation of tau also plays a role in epileptogenesis and ictogenesis in AD. It is reported that the hyper-phosphorylation of tau is implicated in the development of seizures in AD model [63] while reduction of tau or tau phosphorylation supresses seizure and impedes the development of epilepsy [64,65,66,67]. This body of evidence suggests a mechanistic relationship between tau phosphorylation and development of seizures and epilepsy. It is likely that GSK-3, as one of the main kinases of tau, plays a critical role in this process. Furthermore, studies suggested that GSK-3 might be a downstream regulator of other tau kinases such as cyclin-dependent kinase 5 (CDK5) as well as protein phosphatase 2A (PP2A) [68,69,70], which suggested the potential regulating role of GSK-3 in tau phosphorylation. Therefore, GSK-3 represents an attractive target of intervention in AD-associated seizures, such that inhibiting the activity of GSK-3 might have a therapeutic potential in ameliorating or suppressing epileptic seizures in AD models. Although the therapeutic potential of GSK-3 inhibitor in AD patients has not been clarified, several GSK-3 inhibitors such as AZD1080 and Tideglusib are undergoing clinical trial [71,72].

In addition, studies showed that GSK-3 might be involved in AD-associated seizures through alternate pathways. Multiple studies have shown that GSK-3β can be associated with the activity of Fyn [73,74,75], a tyrosine kinase involved in mediating seizure susceptibility through activation of N-methyl-D-aspartate (NMDA receptors which is an ion channel modulating the influx of calcium [76]. Fyn could phosphorylate a unique site Tyr18 (Y18) of tau [77], and Y18-phosphorylated tau is implicated in tau-related NMDA receptor overactivation and excitotoxicity in vitro [78]. Y18-phosphorylated tau subsequently over-activates NMDA receptors and causes excitotoxicity and eventually seizures. Taken together, these results might lead to another pathway of seizure development that involves GSK-3 in AD (Figure 2). However, it is worth mentioning that evidence in physiological condition is required to prove this hypothesis. Studies in AD animal models are also important to connect the dots among GSK-3, Fyn, and AD.

## 4. Putative Interaction Between APP and GSK-3

Although the interaction between GSK-3 and tau has been widely studied, the mechanisms which initiate the dysregulation of GSK-3 activity potentially leading to subsequent seizures and epileptiform events in AD remain to be clarified. Multiple results from in vitro and in vivo experiments showed that GSK-3β might be activated by APP or its metabolites [57,79,80,81]. Studies found that the C-terminus fragment of APP, including AICD and its downstream cleavage product C31, increased the expression of GSK-3 [82,83]. Furthermore, other study also found the increase of activated GSK-3 in AICD overexpressing mice [84]. These lines of evidence showed that it might be this intracellular cleavage product instead of the extracellular β-amyloid peptides that activates GSK-3β (Figure 1B). However, further research is needed to clarify specifically which metabolite(s) of the APP protein are relevant.

Interestingly, some evidence showed that the AICD may also be pertinent to the occurrence of spontaneous seizures and increased susceptibility of induced seizures in animal models. It is reported that the cleavage of APP at the C-terminus may be related to neuronal network abnormality [34], while other studies suggested that the AICD from C-terminus cleavage is critical to neuronal network excitability, possibly leading to the increased spontaneous seizures in transgenic mice harbouring the human AICD gene [34,85]. Therefore, these studies established that AICD or the pathway that AICD is involved in may modulate neuronal excitability and epileptogenesis. Furthermore, there are some studies mentioning that the excitotoxic effect of AICD was exerted through GSK-3 [82,83] whereas inhibition of GSK-3 abated AD phenotypes in AICD transgenic mice [86], which suggests that GSK-3 might be part of the AICD pathway involved in AD phenotypes. However, if GKS-3 is involved in an AICD-related pathway, the relationship between GSK-3 and AICD is unclear, as the activation of GSK-3 in AICD transgenic mice does not clarify whether GSK-3 is upstream or downstream of AICD. On the one hand, observations that GSK-3 activation occurs after the expression of AICD in vitro [87] and in vivo [84,86] provide some lines of evidence that GSK-3 might be downstream of AICD. On the other hand, another report showed that GSK-3 acts upstream of AICD by regulating the activity of AICD through phosphorylation at site Thr668 (T668), thereby greatly enhancing binding of AICD and Fe65 [83], an adaptor protein that facilitates the functions of AICD [88]. As results from multiple studies showed that the binding between AICD and its binding partner Fe65 is essential for the pathogenic role of AICD in AD and epilepsy [34,86,88], it is possible that the activity of AICD is regulated by GSK-3. However, more research is required to prove the causal relationship between AICD and GSK-3, as direct evidence is still missing to clarify whether GSK-3 activity is involved in the development of seizures in AICD transgenic mice. It is possible that activation of GSK-3 and the development of seizures are two independent events in AICD transgenic mice. Therefore, further study is necessary to support the role of GSK-3 in epileptogenesis and ictogenesis in AD.

Some other studies suggest that GSK-3 might also be involved in the production of AICD through the cleavage of APP [55,89]. Phiel and colleagues showed that GSK-3 accelerated the cleavage of APP and the production of C-terminus fragments, whereas inhibition of GSK-3 reduces the production of C-terminus fragment [90]. Therefore, as the increased production of AICD is correlated to the development of seizures and epilepsy [34,85], it is possible that GSK-3 is implicated in epilepsy development through increasing the production of AICD. However, in what way GSK-3 is involved in the processing of APP has not been clarified to date. Although it has been reported that suppressing the expression of GSK-3α after birth of mice reduces the production of C-terminus fragments [90], other study showed that knockout of GSK-3α gene before birth does not alter the processing of APP [89]. This discrepancy can be explained by a compensatory mechanism that both GSK-3α and GSK-3β mediate the processing of APP. While inhibiting GSK-3α acutely decreases the cleavage of APP, the functions of GSK-3α might be overtaken by GSK-3β chronically. However, further research is needed to verify this mechanism.

Collectively, accumulating evidence suggests that there might be a GSK-3-amyloid-tau triad that induces the development of seizures and epilepsy in AD. However, the specific interaction between GSK-3 and AICD remains to be clarified. Answering these questions might help us understand the pathophysiology of AD-associated epilepsy as well as develop new therapeutic approaches to manage seizures and epilepsy in AD patients.

## 5. Alternative Roles of GSK-3 in AD and Seizures

As previously discussed, the occurrence of seizures may accelerate the progress of AD. Some evidence showed that GSK-3 might also be involved in this process. It is clear that GSK-3 plays an important role in the pathogenesis of AD through the phosphorylation of tau [56,91] and that inhibiting GSK-3 decelerates the progression of AD phenotypes in animal models [86]. Therefore, it is possible that seizures aggravate AD phenotypes by increasing GSK-3 activity and hyper-phosphorylated tau. The activation of GSK-3 and phosphorylation of tau have been shown in chemical and electrical induced epilepsy models [54,92,93]. However, the results of GSK-3 activation are not always consistent in post-seizure models. One study showed that GSK-3 was not activated 24 h after kainate-induced seizures [94], which might be explained by the observation that the activation of GSK-3 might last for less than 8 h [95]. Furthermore, some studies showed that there might be some regulatory factors working against the activation of GSK-3 after seizures. It is reported that GSK-3 could not be activated 24 h after seizures, while the activation of GSK-3 lasted longer than 24 h in Dopamine D2 receptor (D2R) knockout mice [94,96]. Interestingly, the loss of D2R was also reported in AD patients [97,98]. Lines of evidence suggest that D2R might suppress the activation of GSK-3 while decrease of D2R releases the suppression and causes the activation of GSK-3. However, the relevance of these results to AD still needs to be considered cautiously, as they were not conducted in AD animal models. More importantly, these studies do not specify whether it is seizures per se that directly activate GSK-3, as it might be the exogenous stimuli such as chemical or electrical stimulations that increase the activity of GSK-3. Therefore, it is too early to draw the conclusion that seizures exacerbate AD through GSK-3.

Based on the current evidences, it may be hypothesized that GSK-3 contributes to the “vicious cycle” between seizure and AD. After being activated in AD, GSK-3 might induce the occurrence of epileptic seizures, which could further increase the activity of GSK-3 and establish a positive feedback mechanism. Therefore, regulation of GSK-3 activity might be a potential intervention for both AD and epileptic seizures.

There are also multiple studies that link GSK-3 to seizures in non-AD models. One study showed that GSK-3β is an essential regulator of the cystine/glutamate antiporter x_c_-, a system that imports cystine and exports glutamate [97]. The disturbance of GSK-3β activity might lead to the spillover of glutamate, which may trigger spontaneous seizures [99]. Inhibition of GSK-3 showed anticonvulsant effect in rodents and zebrafish models of temporal lobe epilepsy [100]. However, some studies suggest that simply decreasing the activity of GSK-3 might not be the appropriate solution. Engel and colleagues showed that both increase and decrease of GSK-3β activity acutely aggravated the development of seizures in acquired seizure model [95]. Furthermore, a recent research showed that modified GSK-3β with increased activity decreased the progression of kainate-induced epileptogenesis [101], which was contradictory to many other results. Therefore, the specific role of GSK-3 in epileptic seizures and AD-associated epileptic seizures has to be further clarified.

## 6. Future Research Directions

There is increasing evidence supporting the cooccurrence of AD and seizures. To improve the understanding of both fields and to develop novel therapeutic approaches targeting AD-associated seizures, there are many questions to be addressed. Firstly, GSK-3 activity in AD models has not been well studied. Although some studies have found that GSK-3 could be activated in transgenic human APP mouse models [79,84], the results need to be interpreted cautiously. Some studies inferred GSK-3 activity by measuring protein levels of either activated GSK-3 (Y216/Y279) or inactivated GSK-3 (S9/S21), which might not necessarily correlate with the enzymatic activity of the protein. Therefore, it is important to accurately measure GSK-3 activity through a quantitative kinase activity assay.

Importantly, whether and at what time point GSK-3 is activated in AD models requires further study. To date, no study has reported the chronological analysis on the activity of GSK-3 in animal models carrying AD-relevant mutations. It is possible that GSK-3 activity is increased at an early age due to the gene mutation or the overexpression of genes, such as APP and AICD. The former is more likely to be relevant to human condition while the latter is limited to animal models, so carefully designed studies need to exclude simple overexpression of APP as a driving factor. In addition, a concern raised by Saito et al. suggested the possibility that some phenotypes in APP-overexpressing mice might come from membrane protein overexpression [102]. Furthermore, it is also likely that GSK-3 activity fluctuates over time in animal models. Notably, some studies showed that unprovoked seizures and epileptiform events could occur in AD transgenic models [14,31]. As epileptic seizures might also disturb the activity of GSK-3, it is possible that the GSK-3 activity also changes with the occurrence of seizures. Therefore, the variables that can change GSK-3 activity in AD have to be considered cautiously. It might be AD-associated seizures instead of AD mutations that induce the activation of GSK-3. Investigating GSK-3 activity at different timepoints with the monitoring of seizures might help us unravel the cause and effect relationship between AD phenotypes and GSK-3 activation.

Second, the roles of GSK-3 in AD-associated seizure need to be clarified. Although GSK-3 has been recognised as one of the contributors of AD pathogenesis, whether GSK-3 causes or accelerates the development of epilepsy in AD has not been well studied. One study reported that genetically modified GSK-3 with increased activity decelerates kainate-induced epileptogenesis [101], which suggests that the role of GSK-3 in epileptogenesis can also be different in AD and non-AD animal models. Furthermore, presuming that GSK-3 contributes to epileptogenesis does not necessarily mean that reducing GSK-3 activity can impede this process. Therefore, inhibiting GSK-3 activity, as most studies proposed, might not be the optimal intervention for AD-associated seizure.

Last but not least, the pathway that involves GSK-3 in AD-associated epileptic seizures requires further investigation. GSK-3 is involved in multiple cell signalling pathways [54], which suggests that GSK-3 might not be implicated directly in the development of seizure and epilepsy. It might be another downstream molecule, such as Fyn and tau, that directly contributes to the onset of seizures, but more evidence is needed to establish the pathway and to verify the therapeutic potential of regulating the targets involved. Therefore, it would be valuable to pinpoint the position of GSK-3 in the pathway that leads to AD-associated seizure.

To summarise, emerging evidence suggests that GSK-3 may play a crucial but complex role in the pathogenesis of epileptic seizures in AD. Unravelling this relationship could potentially open up new therapeutic strategies.

## Figures and Tables

**Figure 1 ijms-21-03676-f001:**
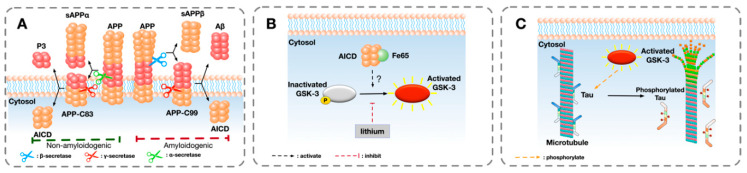
A schematic overview of a hypothesised pathway that GSK-3 is involved in the phosphorylation of tau: (**A**) The sequential cleavage process of amyloid precursor protein (APP) in the amyloidogenic and non-amyloidogenic pathway in Alzheimer’s disease (AD). In the amyloidogenic pathway, the extracellular domain of APP is firstly cleaved by β-secretase, liberating a soluble extracellular N-terminus fragment sAPPβ and a transmembrane fragment AP-C99. The remaining APP-C99 is further cleaved by γ-secretase, generating β-amyloid (Aβ) and releasing APP intracellular domain (AICD) into cytosol. In the non-amyloidogenic pathway, APP is cleaved by α-secretase and γ-secretase, releasing sAPPα, P3, and AICD. (**B**) One possible pathway of the activation of GSK-3. After binding with Fe65, AICD directly or indirectly activates GSK-3 by removing the phosphorylation from serine site (Ser9 of GSK-3β and Ser21 of GSK-3α). The activation of GSK-3 can be inhibited by GSK-3 inhibitor such as lithium. (**C**) GSK-3 phosphorylates microtubule-associated protein tau (MAPT). Tau is involved in the stabilisation of microtubules and axonal transport. After being abnormally phosphorylated by activated GSK-3, hyper-phosphorylated tau detaches from microtubules and causes the dissociation of microtubules.

**Figure 2 ijms-21-03676-f002:**
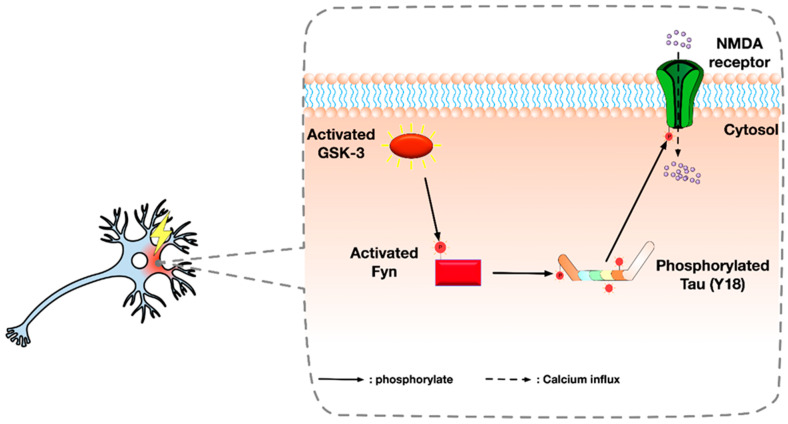
Illustration summarising an alternative pathway of Glycogen synthase kinase-3 (GSK-3): It is possible that GSK-3 activates Fyn by phosphorylation. Activated Fyn then binds to tau and mediates the phosphorylation of tau at tyrosine 18 (Y18). Y18 phosphorylated tau induces the phosphorylation of the NR2 subunit of NMDA receptor at tyrosine 1472 (Y1472) near the C-terminus, which causes the abnormal activation of NMDA receptor and excessive Ca2+ influx and excitotoxicity.

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
