# Peer review of "Unravelling the Role of Glycogen Synthase Kinase-3 in Alzheimer’s Disease-Related Epileptic Seizures"

_ijms, 2020, doi:10.3390/ijms21103676_

Round 1
Reviewer 1 Report
The paper by Runxuan Lin et al is a review on the role of Glycogen Synthase Kinase-3 in Alzheimer’s Disease-Related Epileptic Seizures
GSK-3 might be a key factor that drives epileptogenesis in AD by interacting with myloid precursor protein (APP) and tau, the pathological hallmarks of AD.
In this way, GSK-3 might be involved in initiating a vicious cycle between AD and seizures. The ms is well written and complete. Its importance in the field may help researchers develop new therapeutic approach. Therefore it can be published as it is.
Author Response
We thank the reviewer for their time and positive feedback on our article.
Reviewer 2 Report
The authors address the apparently complicated issue of the role of GSK in AD-related epilepsy in a fairly clear way. A number of studies are summarised and their limitations, as well as discrepancies between them, are nicely discussed. They also propose a plausible and attractive ‘vicious cycle’ hypothesis of GSK activation and AD progression. However, there are a few points that need to be addressed:
In line 297, the authors make a valuable remark to exercise caution in the interpretation of results based on APP-overexpressing AD mouse models. Also, the limited relevance of such models to sporadic AD cases is dutifully noted earlier in the text. Along these lines, it might also be worthwhile to mention the reservations regarding the interpretation of APP-overexpression studies that were raised by Saito et al. J Neurosci. 2016 21;36(38).
It would be informative for non-expert readers to depict non-amyloidogenic processing of APP in Fig. 1A, as a comparison with AD-related cleavage.
It would be nice to visualise the relevance of the problem if some more exact numbers were provided, if available: what percentage of AD patients develop seizures, compared with age-matched controls?
Line 53-54: regarding complex nature of senile plaques (e.g. Armstrong R Folia Neuropath 2009;47:289-299) I would like to encourage authors towards revising the below statement: "There are two main pathological hallmarks in AD, senile plaques and neurofibrillary tangles (NFT) which involve amyloid precursor protein (APP) and microtubule-associated protein tau (MAPT), respectively."
While the message is crucial, lines 79-80 are basically lines 46-47 barely rewritten (plus typo ,,controlS” in line 47).
Line 219: no explanation of what Fe65 is in the sentence that first mentions it.
Moreover, some language issues need to be revised:
Line 195: ‘remain’, instead of ‘remains’
Unnecessary capital letter in ‘lithium’ in Fig.1 caption.
Style: the word ‘relevant’ repeated 3 times between lines 201-205; ‘further study is’ repeated in lines 221 and 225; ‘compensatory’, ‘compensated’ and ‘compensatory’ repeated throughout lines 236-239; ‘these lines of evidence’ repeated in lines 259 and 262, plus typo in line 259 (‘there’ instead of ‘these’)
Line 209: ‘...AICD or the pathway that AICD is involved IN may modulate...’
Lines 214 and 216: unnecessary ‘the’ before ‘downstream/upstream’
Line 216: ‘observations (…) provide’
Lines 223, 241, 306, 320: ‘seizureS’
Line 260: ‘causeS’
